A comparison of two gluteus maximus EMG maximum voluntary isometric contraction positions

Contreras Bret 1
Vigotsky Andrew D. 2 avigotsky@gmail.com
Schoenfeld Brad J. 3
Beardsley Chris 4
Cronin John 1 5
1 Auckland University of Technology, Sport Performance Research Institute New Zealand , Auckland , New Zealand
2 Kinesiology Program, Arizona State University , Phoenix, AZ , USA
3 Department of Health Sciences, CUNY Lehman College , Bronx, NY , USA
4 Strength and Conditioning Research Limited , London , UK
5 School of Exercise, Biomedical and Health Science, Edith Cowan University , Perth , Australia
Reser David
Electronic publication date: 2015 Sep 22
Publication date: 2015
Volume: 3
Electronic Location ID: e1261
Received 2015 Aug 14; Accepted 2015 Sep 2
Copyright: © 2015 Contreras et al.
Copyright year: 2015
Copyright holder: Contreras et al.
License: This is an open access article distributed under the terms of the Creative Commons Attribution License, which permits unrestricted use, distribution, reproduction and adaptation in any medium and for any purpose provided that it is properly attributed. For attribution, the original author(s), title, publication source (PeerJ) and either DOI or URL of the article must be cited.
License URL: https://creativecommons.org/licenses/by/4.0/

Keywords: MVC, MVIC, Electromyography, Neuromechanics, Normalization

Funding: The authors received no funding for this work.

==============================
Background. The purpose of this study was to compare the peak electromyography (EMG) of the most commonly-used position in the literature, the prone bent-leg (90°) hip extension against manual resistance applied to the distal thigh (PRONE), to a novel position, the standing glute squeeze (SQUEEZE).

Methods. Surface EMG electrodes were placed on the upper and lower gluteus maximus of thirteen recreationally active females (age = 28.9 years; height = 164 cm; body mass = 58.2 kg), before three maximum voluntary isometric contraction (MVIC) trials for each position were obtained in a randomized, counterbalanced fashion.

Results. No statistically significant (p < 0.05) differences were observed between PRONE (upper: 91.94%; lower: 94.52%) and SQUEEZE (upper: 92.04%; lower: 85.12%) for both the upper and lower gluteus maximus. Neither the PRONE nor SQUEEZE was more effective between all subjects.

Conclusions. In agreement with other studies, no single testing position is ideal for every participant. Therefore, it is recommended that investigators employ multiple MVIC positions, when possible, to ensure accuracy. Future research should investigate a variety of gluteus maximus MVIC positions in heterogeneous samples.

Introduction

Maximum voluntary isometric contractions (MVIC) are often used to normalize electromyography (EMG) signals. It is important to employ an MVIC position that elicits the highest activation in order to increase the validity of EMG studies and decrease incidents of abnormally high normalized mean and peak EMG data. In order for accurate comparisons to be made between studies, it is also important for researchers to standardize MVIC positions, or at least use positions that elicit similar magnitudes of EMG activity. A number of MVIC positions have been used in the literature to assess the gluteus maximus, including the Biering-Sorenson position (Cambridge et al., 2012; McGill, McDermott & Fenwick, 2009), the prone straight leg hip extension position (Barton et al., 2014; Worrell et al., 2001), the prone bent leg position (Jakobsen et al., 2013; Youdas et al., 2013), the prone straight leg position with 70° of hip flexion (Simenz et al., 2012), and the standing bent leg position (Boudreau et al., 2009). The most commonly used position, however, is the prone bent-leg (90°) hip extension with manual resistance applied to the distal thigh (PRONE) (Choi et al., 2015; Emami, Arab & Ghamkhar, 2014; Hislop et al., 2014; Kang et al., 2013; Kendall , 2005; Oh et al., 2007).

A recent study by Simenz et al. (2012) that used a prone gluteus maximus MVIC position in 70° of hip flexion, demonstrates the importance of standardizing MVIC positions across studies. Researchers have shown that lower gluteus maximus amplitude is elicited at higher degrees of hip flexion and reaches a maximum EMG amplitude at end range hip extension (Worrell et al., 2001). By employing an MVIC position that renders significantly lower EMG activity than those values that are truly maximal, the normalized data of Simenz et al. (2012) are most likely overestimated. For example, if the work of Worrell et al. (2001) is extrapolated, the MVIC position used by Simenz et al. (2012) would only elicit approximately 80% of true MVIC, translating into 25% greater mean and peak values when compared to the true MVIC position. The data reported by Simenz et al. (2012) therefore cannot be used for comparison with exercises in other studies that utilized alternative MVIC positions with smaller hip flexion angles, as the data would have overestimated how effectively the gluteus maximus was activated. Therefore, it is apparent that researchers should only compare EMG data that utilize positions that render similar values.

Since Worrell et al. (2001) found that full hip extension elicited the greatest amount of gluteus maximus EMG activity, and this finding is corroborated by earlier work from Wheatley & Jahnke (1951) and Fischer & Houtz (1968), it is postulated that the most appropriate gluteus maximus MVIC position is at full hip extension, or hip hyperextension. PRONE is currently the recommended position in several texts on muscle testing (Hislop et al., 2014; Kendall , 2005), although to the authors’ knowledge, this position has not been compared to others in the literature. In order to correct for individual variation, some researchers have employed multiple MVIC positions. For example, McGill, McDermott & Fenwick (2009) used both the Biering-Sorenson and PRONE positions; whichever position elicited the greatest activity was used for normalization purposes. The authors, however, are unaware of any existing research that quantitatively compares gluteus maximus MVIC positions.

The gluteus maximus muscle appears to be segmented into at least two subdivisions, which may display different EMG activity in response to certain muscle actions. McAndrew, Gorelick & Brown (2006) used a laser-based mechanomyographic (MMG) technique to measure the mean contraction time in six subdivisions of the gluteus maximus, both in the sagittal plane (superior, middle, inferior) and in the frontal plane (medial and lateral). The superior region displayed the longest contraction time followed by the middle region and then the inferior region. On the basis of these findings, McAndrew, Gorelick & Brown (2006) suggested that the superior region may contain more slow twitch fibers and be more involved in postural tasks compared to the inferior region, while the inferior region may contain more fast twitch fibers and be more involved in dynamic tasks. This is further substantiated by the work of Lyons et al. (1983) and Karlsson & Jonsson (1965), who found differences between upper and lower gluteus maximus EMG during functional movement; for example, load acceptance during stair ambulation better targets the lower gluteus maximus (Lyons et al., 1983), while hip abduction better targets the upper gluteus maximus (Karlsson & Jonsson, 1965).

Pilot data from our lab showed that some subjects were able to elicit greater EMG activity during a standing glute squeeze (SQUEEZE) when compared to PRONE, and this was especially true for the upper gluteus maximus. Given this observation and the findings articulated in previous paragraphs, the purpose of this investigation was to compare upper and lower gluteus maximus EMG activity in PRONE versus SQUEEZE. Based on our pilot data, it was hypothesized that SQUEEZE would elicit greater upper gluteus maximus EMG activity, while PRONE would elicit greater lower gluteus maximus EMG activity.

Methods

Subjects

Thirteen healthy women (age = 28.9 ± 5.1 years; height = 164 ± 6.3 cm; body mass = 58.2 ± 6.4 kg) with 7.0 ± 5.8 years of resistance training experience participated in this study. Inclusion criteria required subjects to be between 20 and 40 years of age and have at least 3 years of consistent resistance training experience. All subjects were healthy and free of any musculoskeletal or neuromuscular injuries, pain, or illnesses. Subjects completed an Informed Consent form. Subjects were advised to refrain from training their lower body for 72 h prior to testing. The study was approved by the Auckland University of Technology Ethics Committee (AUTEC Reference number 13/375).

Procedures

Subjects first performed a 10-minute general warm-up consisting of various dynamic stretches for the lower body musculature. Following warm-up, subjects practiced each testing position several times, until they felt comfortable with the technique. Subjects were asked to wear appropriate clothing for access to the EMG electrode placement sites. Before placing the electrodes on the skin, excess hair was removed with a razor, and skin was cleaned and abraded using an alcohol swab. After preparation, self-adhesive disposable silver/silver chloride pre-gelled dual snap surface bipolar electrodes (Noraxon Product #272, Noraxon USA Inc., Scottsdale, AZ) with a diameter of 1 centimeter (cm) and an inter-electrode distance of 2 cm were attached in parallel to the fibers of the right upper and lower gluteus maximus. More specifically, “[upper gluteus maximus] electrodes were placed two finger’s width above the line just under the spina iliaca posterior superior and the trochanter major; [lower gluteus maximus] electrodes were set below the same line” (Fujisawa et al., 2014) (Figs. 1 and 2). After the electrodes were secured, a quality check was performed to ensure EMG signal validity.

Figure 1 Prone bent-leg hip extension against manual resistance.

Manual resistance is applied to the distal thigh while the subject generates a hip extension moment.

Figure 2 Standing glute squeeze.

The subject squeezes her glutei, which generates hip extension and external rotation moments.

Following electrode placement, subjects completed three trials of PRONE then SQUEEZE, or vice versa. The PRONE position was performed by having the subject lie prone on the bench, flexing their knee to 90°, and was told to extend her hip while manual resistance was applied to the distal thigh (Fig. 1). The SQUEEZE position was performed by having the subject stand with her feet slightly wider than shoulder width apart and hips slightly externally rotated. The subject was instructed to squeeze her glutei and focus on externally rotating and extending her hips (Fig. 2). For example, if a subject was randomized to complete PRONE first, her testing order would be PRONE, SQUEEZE, rest, PRONE, SQUEEZE, rest, PRONE, SQUEEZE. Each contraction phase lasted 5 s, and each rest phase lasted 3 min. Randomization was counterbalanced so that half the subjects performed PRONE first and the other half performed SQUEEZE first. In all MVIC positions, subjects were instructed to contract the gluteus maximus “as hard as possible.”

Raw EMG signals were collected at 2,000 Hz by a Myotrace 400 EMG unit (Noraxon USA Inc., Scottsdale, AZ, USA). Data was sent in real time to a computer via Bluetooth and recorded and analyzed by MyoResearch 3.6 Clinical Applications software (Noraxon USA, Inc., Scottsdale, AZ, USA). A 10–500 Hz bandpass filter was applied to EMG data. Signals of all MVIC trials were full-wave rectified and smoothed with a root mean square (RMS) algorithm with a 100 ms window. Maximal peak EMG values over a 1,000 ms window, or the 1,000 ms window with the greatest average EMG amplitude within the 5 s contraction, were then used to normalize peak EMG signals obtained during each MVIC trial (Vera-Garcia, Moreside & McGill, 2010).

Statistical analysis

Paired samples t-tests were performed after checking normality using Shapiro–Wilk test in Stata 13 (StataCorp LP, College Station, TX). Alpha was set a priori at 0.05 for statistical significance. Effect sizes (ES) were calculated by Cohen’s d using the formula d=Mdsd, where Md is the mean of differences and sd is the standard deviation of differences (Becker, 1988; Morris, 2008; Smith & Beretvas, 2009). This method is slightly different than the traditional method of calculating Cohen’s d, as it calculates the within-subject ES rather than group or between-subject ES. ES were defined as small (0.20–0.49), moderate (0.50–0.79), and large (⩾0.80) (Cohen, 1988). Confidence intervals (95% CI) for each ES were also calculated.

Results

The normalized peak EMG for the different exercises and gluteus maximus sections can be observed in Table 1. In terms of the upper gluteus maximus comparison, no significant differences were observed in the peak EMG for both exercises (ES = 0.005; 95% CI [−0.599–0.609]; t(12) = 0.018; p = 0.986). With regards to the lower gluteus maximus, a small ES was observed (0.412; 95% CI [−0.193–0.102]) between the two positions in favor of the PRONE position; however, this outcome may have been due to chance alone (t(12) = −1.485; p = 0.164).

Table 1 Group mean ± SD of normalized peak EMG amplitudes.

	Prone	Squeeze	
Upper gluteus maximus	91.94 ± 11.64	92.04 ± 11.30	
Lower gluteus maximus	94.52 ± 13.59	85.12 ± 12.64	

Discussion

The purpose of this investigation was to compare a novel gluteus maximus MVIC position, SQUEEZE, to the current gold standard, PRONE. We have failed to reject the null hypotheses, as there were no statistically significant differences between the two positions tested (Table 1). However, despite no statistically significant differences, the peak EMG values for the lower gluteus maximus were approximately 9% higher for the PRONE compared to the SQUEEZE. Consequently, if the SQUEEZE test were used for normalization, it would render approximately 10% higher mean and peak EMG values compared to the PRONE test. Therefore, although not statistically significant, the findings could be considered practically meaningful. Furthermore, these data show a large amount of individual variation (Table 2), which has been previously described by McGill (1990) and Vera-Garcia, Moreside & McGill (2010) for other muscles.

Table 2 Number of subjects (percentage of subjects (%)) which each MVIC technique resulted in the greatest peak EMG amplitude.

	Prone	Squeeze	
Upper gluteus maximus	7 (53.85)	6 (46.15)	
Lower gluteus maximus	10 (76.92)	3 (23.08)	

There are several kinematic and kinetic differences between PRONE and SQUEEZE, any of which may have affected our results, either individually or in combination. During PRONE, the knee is bent to 90°, whereas during SQUEEZE, the knees are fully extended. Previous research has shown that gluteus maximus EMG activity during hip extension is greater with the knees flexed than when extended, presumably resulting from a greater reliance upon the gluteus maximus for hip extension due to decreased hamstrings length (Kwon & Lee, 2013). On the other hand, extended knees allow for greater hip extension range of motion compared to flexed knees, thereby shortening the gluteal fibers to a greater extent (Van Dillen et al., 2000) and leading to a greater amount of gluteus maximus EMG activity (Worrell et al., 2001). In addition, PRONE involved primarily hip hyperextension since the pelvis was fixed, whereas SQUEEZE appeared to involve a combination of hip extension and posterior pelvic tilt. Although posterior pelvic tilt mimics hip extension (Neumann, 2010), it is unclear how each of these kinematic variables might affect gluteus maximus EMG activity individually. To our knowledge, no study to date has investigated gluteus maximus EMG activity with varying combinations of hip extension and posterior pelvic tilt during MVIC actions. Moreover, PRONE is an open kinetic chain maneuver with the torso stabilized onto a bench, whereas SQUEEZE is a closed kinetic chain maneuver performed in a standing position. Stensdotter et al. (2003) investigated the EMG activity of the quadriceps muscle group during open kinetic chain and closed kinetic chain positions during MVIC actions and reported significant differences in EMG amplitude. The rectus femoris displayed greater EMG activity during open kinetic chain maneuvers while the vastus medialis displayed greater EMG activity during closed kinetic chain maneuvers. It is therefore hard to predict whether the gluteus maximus would inherently display greater or lesser EMG activity during either open or closed kinetic chain maneuvers. Finally, PRONE required manual resistance, whereas SQUEEZE relied upon anatomical structures surrounding the hip to provide resistance against hip extension. Whether this factor has any effect on EMG activity recorded in a muscle is unclear, as the authors are unaware of any previous investigations into the effect of squeezing a muscle whereby range of motion is limited by anatomical structures on EMG activity rather than against external resistance.

This investigation was subject to several important limitations. Firstly, although we observed what may have been a practically important difference between the MVIC positions, this difference was not found to be statistically significant, which suggests that our initial estimates for the appropriate sample size may have been too small. Secondly, there were several kinematic differences between the two positions that were explored (PRONE and SQUEEZE), including different pelvic, hip, and knee joint angles. There were also kinetic differences between the two positions, in that PRONE was an open kinetic chain maneuver and SQUEEZE was a closed kinetic chain maneuver. Moreover, PRONE used external resistance and SQUEEZE utilized oppositional torques produced by internal, anatomical structures. These multiple differences make it difficult to assess whether our results arose from a combination of biomechanical factors acting in opposing directions, heterogeneity, or genuinely no difference between the conditions. Thirdly, we only compared two MVIC positions, and it is feasible that other positions might result in superior or inferior levels of EMG activity. Fourthly, we only investigated two subdivisions of the gluteus maximus muscle and there are indications that there may be others, from proximal-to-distal, medial-to-lateral, and superficial-to-deep. Furthermore, our statistical analysis was not designed to assess whether there was a difference between the EMG activity of the upper and lower gluteus maximus during either MVIC position and therefore this remains uncertain.

Conclusions

Although these data are inconclusive as to which position is superior, they do provide insight as to the complexity of MVIC positions for the gluteus maximus. More specifically, due to the large individual variations (Table 2), it is recommended that multiple MVIC positions be utilized to ensure that the greatest possible EMG amplitude be the divisor during normalization. These recommendations are well in line with other studies, which have utilized or recommended multiple MVIC positions (McGill, McDermott & Fenwick, 2009; Vera-Garcia, Moreside & McGill, 2010). Future research should use heterogeneous samples, such as athletic males, and also test more positions, such as the Biering-Sorenson position, quadruped hip extension position, and top hip thrust position (Contreras, Cronin & Schoenfeld, 2011), each with manual resistance, along with the tall kneeling position.

Supplemental Information

Supplemental Information 1 Raw data

Click here for additional data file.

Additional Information and Declarations

Competing Interests

Author Contributions

Human Ethics

Chris Beardsley is the founder and owner of Strength and Conditioning Research Limited. All other authors declare that they have no competing interests.

Bret Contreras conceived and designed the experiments, performed the experiments, contributed reagents/materials/analysis tools, wrote the paper, reviewed drafts of the paper.

Andrew D. Vigotsky performed the experiments, analyzed the data, wrote the paper, prepared figures and/or tables, reviewed drafts of the paper.

Brad J. Schoenfeld, Chris Beardsley and John Cronin wrote the paper, reviewed drafts of the paper.

The following information was supplied relating to ethical approvals (i.e., approving body and any reference numbers):

Auckland University of Technology Ethics Committee

AUTEC Reference number 13/375.

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
