# Peer review of "A comparison of two gluteus maximus EMG maximum voluntary isometric contraction positions"

_PeerJ, doi:10.7717/peerj.1261_

## Round 0.1 · original submission · Minor Revisions

Both reviewers were highly complimentary regarding the study design and quality of data presentation, and both also made specific recommendations that will improve the final presentation. In particular, I agree that a schematic or drawing illustrating the body positions tested, as well as clarification of the selection method for the sampled interval will benefit the readers of the published version. Please carefully examine and address the points raised by the reviewers, and I look forward to reading your revised manuscript. Thank you again for submitting your work to PeerJ.

·

Basic reporting

I feel that the authors have done a nice job on this study. The manuscript is well-written. I personally enjoy these methodological studies because as the authors have mentioned, the interpretation of data can vary widely depending on the choices and decisions of the research team.

No Comments.

Experimental design

The methodological approach and statistical analyses used in this study were appropriate. My only recommendation would have been to allow a separate familiarization session to practice the techniques on a separate day prior to data collection to alleviate any concerns regarding accumulation of muscle fatigue. In addition, I think the paper would benefit by describing to the reader precisely how it was decided which 1,000 ms epoch was used for the amplitude analyses. My assumption is that these signals were fairly stationary, but it would be helpful for future investigators to know this.

Validity of the findings

The findings of this study seem valid and the statistical approach was sound. The use of effect sizes and 95% confidence intervals was a welcomed addition. The limitations paragraph of the Discussion section was helpful for readers to truly appreciate the complexity of EMG studies in motor control and biomechanics.

No major comments

Additional comments

I do have some minor comments and recommendations. These are most based on personal preference and the authors can proceed as they wish.

1. I would consider minimizing the use of abbreviations. Terms such as EMG and MVC are obviously very common, but abbreviations such as GM, UGM, and LGM become confusing and are generally not helpful. In addition, I don't see a need for prone to be capitalized throughout. I found this to be a bit annoying.
2. Line 60: Using the word "degrees" multiple times may lead to confusion.
3. Line 65, pg. 2: A year should be included with the Simenz reference.
4. Line 158: Typically effect sizes are reported as absolute values. Obviously, the order is which the means are subtracted will dictate whether the outcome is positive or negative, but I think it makes more sense to provide the absolute value, particularly because the means have been displayed for the reader in Table 1.
5. Line 170: As the authors are aware, the term "reject the null hypothesis" is often used in statistical jargon to describe a true difference between means. The authors use of "rejected" in line 170 has the potential to lead to some confusion, since the rejection of this hypothesis is the opposite of rejection of the null hypothesis. I would consider rewording this.

Reviewer 2 ·

Basic reporting

I found the paper extremely well written, so well done, it is a pleasure to read an article that does not have any grammatical errors.

Experimental design

The experimental design was simple and easy to understand what the authors wanted to achieve. One point though is that it could be made clearer the positions that you tested with a diagram so I would like that to be included for publication to go ahead.

Validity of the findings

The findings of no differences in the two MVIC positions is interesting as I would of thought also that the squeeze may have elicited more EMG. The more studies done on MVC positions the better which will lead to better standardisation between studies so this study is valid in that it tries to achieve this.

Additional comments

they need to put in some diagrams of their two MVIC positions just so the readers can be exactly clear on these two positions.

---

## Round 0.2 · Minor Revisions

I believe that the revisions suggested by the reviewers have improved the overall readability of the manuscript, and its utility for researchers in the field. I have only one minor additional suggestion, which would be to add markings (e.g. circles or arrowheads) to figures 1 and 2 indicating the positions of the recording electrodes used during the EMG trials. I believe this will aid future investigators when considering the effects of possible functional subdivisions of the muscle. Apart from that, the manuscript is suitable for publication, and I look forward to seeing it in print. Thank you again for choosing PeerJ as a forum for your work.

---

## Round 0.3 · accepted · Accept

Thank you for adding those figure elements. I look forward to seeing your paper online.